# Backup power supply for a hydrogen-producing offshore wind turbine - a technology comparison

Linus Niklaus[1], Paul Rothe[1], Johannes Liebertseder[1], and Martin Doppelbauer[1,2]

[1]Fraunhofer Institute for Chemical Technology ICT, New Drive Systems, 76131 Karlsruhe, Germany
[2]Karlsruher Institue of Technology (KIT), Institute of Electrical Engineering (ETI), 76131 Karlsruhe, Germany

**Correspondence:** Linus Niklaus (linus.niklaus@ict.fraunhofer.de)

**Abstract.** Hydrogen is an important energy carrier for the transition of the energy sector towards decarbonization. An attractive option for its sustainable production are independent offshore wind farms that include all systems for the hydrogen production directly on the wind turbine platform. However, these systems face a challenge in maintaining a constant electric power supply on the turbine platform during periods of calm winds as they are not connected to the onshore electrical grid. This study evaluates and compares different technologies for a backup power supply on the wind turbine platform. Due to the limited energy storage capacities of battery systems and thus, short energy bridging times, systems are investigated that make use of the platform-produced hydrogen to generate electricity and consequently provide long bridging times. Three different backup power supply systems are investigated: A fuel cell system combined with a battery storage system (H2-FC+BS) as well as a hydrogen internal combustion engine with and without a battery storage system (H2-ICE+BS and H2-ICE). These systems are examined in terms of efficiency (hydrogen consumption), lifetime, robustness, maintenance requirements, space consumption, and costs. The results suggest that the hybrid system of a hydrogen combustion engine with an accompanying battery storage unit provides an optimal solution, offering a balanced compromise between efficiency, robustness, and minimized maintenance demands.

## 1 Introduction

Offshore wind energy has great potential for the production of green hydrogen but is also very challenging (Rodríguez Castillo et al., 2024). The demand of environmentally friendly hydrogen is constantly increasing as part of the industrial transition to climate-neutral technologies. Until now, it has been common practice to transmit the electrical energy generated offshore to land via converter stations and submarine cables, where it is either distributed via the electrical grid or used directly for hydrogen production. Because of this very expensive connection, the H2Mare project is investigating how hydrogen can be produced directly offshore. A decentralized solution promises the greatest potential (Rogeau et al., 2023). So each wind turbine is to be equipped with a platform that has all the necessary systems for hydrogen production, such as a desalination plant, the electrolyzer and a gas treatment plant. These subsystems are housed in individual containers that can be manufactured onshore and only need to be connected together on the platform. This results in a flexible and modular system that is independent of the size of the wind farm. Mass production of the system and components is expected to result in a cost advantage over a

centralized platform. The hydrogen can then either be transported onshore via a pipeline or stored offshore and exported by ship. In addition to a reduction in production costs, this is expected to enable new locations for offshore wind farms to be developed in an economically viable manner. However, this also brings new challenges. Normally, wind turbines are supplied with electricity via their grid connection during windless periods and thus kept operational. If this connection is lost, a backup power supply must be created to protect the wind turbine and the hydrogen-producing systems against calm wind. Without such a backup power supply, all systems would fail completely if the turbine came to a standstill. The wind turbine would not be able to restart as there would be no energy available. Furthermore, the systems would run the risk of being damaged by frost in cold temperatures. A robust and reliable backup power supply is therefore one of the most important components for such a system. For this purpose a battery storage system would have to be very large in order to be able to bridge long calm wind. Converting the hydrogen produced back into electrical energy is a more practical option. There are two possible technologies for this. On the one hand, the hydrogen can be converted directly back into electrical energy with the help of a fuel cell. On the other hand, it is possible to use the hydrogen to operate an internal combustion engine and drive a generator.

**Proceeding**

The two competing technologies are compared below for use as a backup power supply under the conditions of the German North Sea to identify the most suitable one. Until now, similar comparisons have been made primarily for mobility applications, as in Mayr et al. (2021) for heavy utility vehicles. For this purpose wind and temperature data from the German North Sea are first evaluated in Section 2. Based on this data and other given boundary conditions, the operating conditions of such a unit are described. Since no hydrogen combustion engines are yet available on the market that would be suitable for this application, a detailed analysis of this component is carried out in Section 3. Section 4 then shows possible system designs and explains their individual components. Simulation models were built on the basis of these systems in order to analyze the efficiency and estimate the annual hydrogen consumption. The results of these observations are presented in section 5, along with other qualitative comparison criteria.

## 2 Analysis of load profile and boundary conditions

There are a lot of components on the platform that have to be operated even when there is no wind. On the one hand, these are systems that are necessary to keep the wind turbine ready for operation so that it can resume work as soon as sufficient wind is available again, such as the nacelle and blade adjustments and control units. On the other hand, hydrogen production on the additional platform must also be kept ready for operation, for which auxiliary operations such as pumps and control and safety systems are required. A constant base load of $7\,\mathrm{kW}$ is assumed for all these systems. In addition, critical components must be heated at cold temperatures to protect them from damage caused by frost. This increases the power requirement of the platform from temperatures below $5\,^{\circ}\mathrm{C}$. The minimum outside temperature that the platform must be able to withstand is $-20\,^{\circ}\mathrm{C}$. The total power requirement here is $50\,\mathrm{kW}$, which corresponds to the required maximum output of the back-up power supply. The increase in power between these two temperatures can be assumed to be linear. This load profile was specified by the project

framework and is based on information provided by the turbine manufacturer and thermal estimates of the container modules to maintain a minimum temperature of $5\,°C$.

The back-up power unit always takes over the supply to the system when the wind turbine is unable to produce electricity, which is mainly the case when the wind speed becomes too low. Below $4.3\,\mathrm{m\,s^{-1}}$, the wind turbine is switched off for economic and technical reasons (Peters and Drillet, 2023). However, the wind turbine cannot be operated even in stormy conditions with wind speeds of over $25\,\mathrm{m\,s^{-1}}$ (Bundesverband WindEnergie e.V. (BWE), 2022) or during maintenance work.

In order to estimate the resulting standstill time and the power requirements of the platform for a location in the German Bight on the basis of the above values, measurement data from FINO1 was evaluated. This is a research platform specifically for offshore wind turbines, which is operated by the FuE-Zentrum FH Kiel GmbH and whose measurement data is made available every 10 minutes from the Federal Maritime and Hydrographic Agency (BSH) (FuE-Zentrum FH Kiel GmbH, 2024).

**Table 1.** Standstill time in 2021 and 2022 grouped by duration

| Standstill time | Frequency | | Total Duration | |
|:---:|:---:|:---:|:---:|:---:|
| | 2021 | 2022 | 2021 | 2022 |
| <= 10 min | 362 | 312 | 60.33 h | 52 h |
| 20 min - 3 h | 580 | 550 | 503.83 h | 511.67 h |
| 3 h - 12 h | 129 | 94 | 821.5 h | 573.33 h |
| > 12 h | 34 | 32 | 709.33 h | 666.5 h |
| **Total** | **1105** | **988** | **2095 h** | **1803.5 h** |

In Table 1, the standstill times of the wind turbine are shown for the years 2021 and 2022 based on the measurement data of FINO1. There are a lot of short standstills but the majority of the total duration in a year results from few long standstills. In 2021, there would have been a total of 1105 interruptions to the energy supply due to the standstill of the wind turbine with a total duration of $2095\,h$. This corresponds to around 3 standstills per day, each with an average length of $1.9\,h$. The 988 interruptions in 2022 result in a total standstill time of $1803.5\,h$, which corresponds to 2.7 standstills per day with an average duration of approx. $1.8\,h$. So, the clam wind periods show the same characteristics in both years.

However, as the temperature also plays a role in addition to the duration and frequency of calm wind periods for the emergency power supply, it is not only the absolute figures for a year that are relevant, but also their seasonal distribution. Therefore, in Figure 1 the monthly distribution of calm wind periods are shown for the years 2021 and 2022. A seasonal dependency of calm wind periods can be observed. In the warm half-year from April to September the standstill time of the turbine is about twice as long as in the winter half. This indicates that colder temperatures, which are to be expected in these months in particular, lose some of their significance in advance.

The German Bight is not known for its cold climate either. In Figure 2, the temperature distribution of the years 2021 and 2022 are shown. In 2021 only 21% of the year the temperature is below $5\,°C$ and the minimal temperature is $-3.3\,°C$, so that an additional heating is rarely needed and only low heat outputs are to be expected. In 2022 these effects are even more pronounced because of the very mild winter. Here the outdoor temperature is for only 11% of the time below the critical value

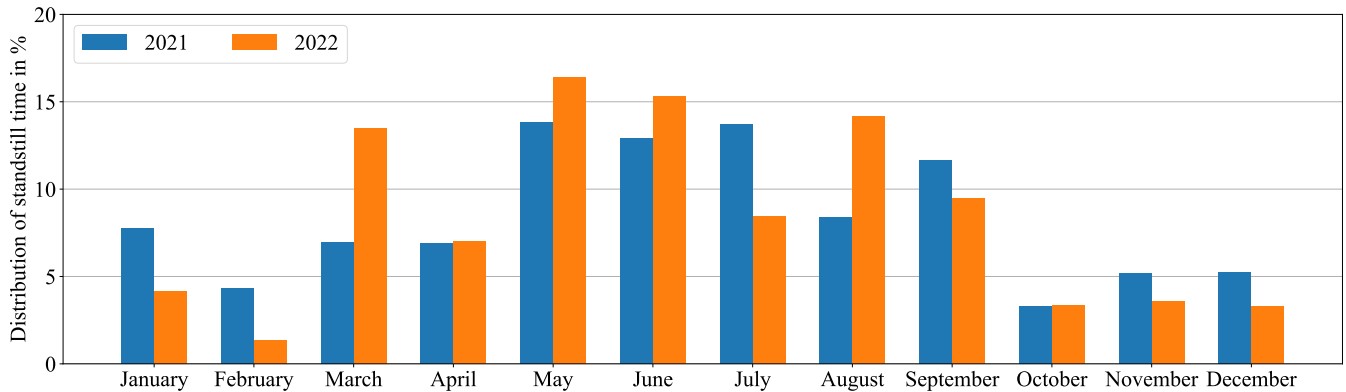

**Figure 1.** Distribution of calm wind periods in 2021 and 2022

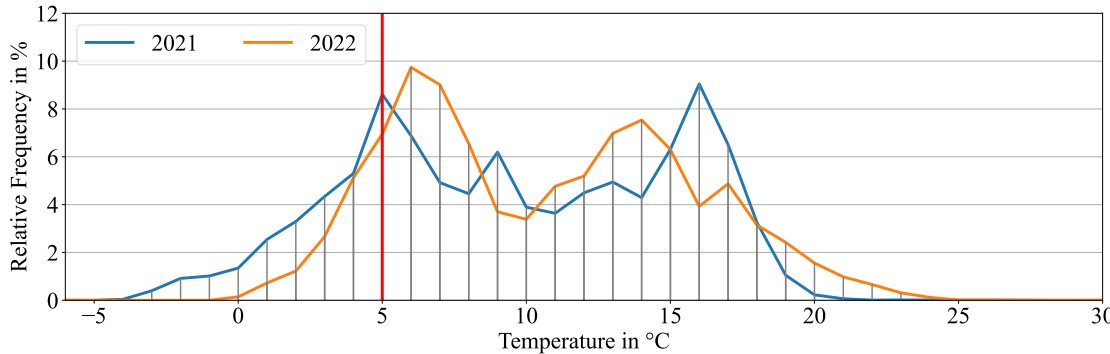

**Figure 2.** Temperature distribution in 2021 and 2022

of $5\,°\mathrm{C}$ and the minimal measured temperature is $0.13\,°\mathrm{C}$. Nevertheless, the unit must still be able to provide a output power of $50\,\mathrm{kW}$ to reliably protect the platform from frost, as other locations in colder regions may also be considered in the future.

Based on the findings described above, four comparative scenarios were derived with corresponding weighting. This makes it possible to estimate the annual hydrogen consumption without having to fully simulate the years under consideration. This significantly reduced the simulation effort. The four scenarios differ in terms of their duration and the required output power, which is associated with different outside temperatures. The standstill time is therefore divided into a short ($20\,\mathrm{min}$) and a long ($6\,\mathrm{h}$) scenario. Similarly, a distinction is made between a low ($7\,\mathrm{kW}$; base load) and a high ($25\,\mathrm{kW}$; $-5\,°\mathrm{C}$ outside temperature)

power requirement. The resulting four scenarios were weighted in relation to each other for each year under consideration so that the actual mean value is calculated month by month. The months were then weighted according to the shares of the total standstill time shown in Figure 1. The resulting weightings are displayed in Table 2. As can be seen, the scenarios with high output power in 2022 lose all significance, as the winter was very mild and correspondingly cold temperatures were very rare.

**Table 2.** Weighting of the comparison scenarios

| | | Standstill time | |
|---|---|---|---|
| | | **short** (20 min) | **long** (6 h) |
| **Power Requirement** | **low** (7 kW) | **Scenario 1** | **Scenario 2** |
| | | 2021: 56.74% | 2021: 26.45% |
| | | 2022: 69.03% | 2022: 30.37% |
| | **high** (25 kW) | **Scenario 3** | **Scenario 4** |
| | | 2021: 13.23% | 2021: 3.59% |
| | | 2022: 0.47% | 2022: 0.13% |

## 3    Detailed design of a hydrogen ICE

The GT-Suite simulation software is used to model the hydrogen combustion engine. In order to validate charge exchange and reaction kinetics and to calibrate the simulation models, a single-cylinder test engine is first modeled and compared with measurement data, which were carried out at the Fraunhofer ICT. The research engine and various series of measurements have already been described in several publications (Bucherer et al., 2023; Gal et al., 2023). The validated single-cylinder simulation model is then extended to a four-cylinder engine. The intake air and exhaust gas paths are optimized and different

turbocharging systems are investigated. Finally, the engine model is applied for certain operating points, whereby the valve lift curve, injection and ignition are adjusted.

When parameterizing the 1D ICE model using experimental measurement data for hydrogen, the modeling of the intake air path, deflagration barrier and valves is first adapted. The model is shown in Figure 3.

In Figure 3 the intake (1) starts at the top left, after which the charge air flows through a throttle valve (2), which is always

open in this configuration. The following calming volume (3) serves to dampen pressure pulsations caused by the gas exchange. The volume was modeled using a network of pipe sections and flanges. Behind this is a deflagration barrier (4) to interrupt any re-ignition of the hydrogen-air mixture. Both the research engine and the final engine are operated via intake manifold injection (5). The mixture finally passes through the intake ducts (6) into the cylinder (7) shown in the middle. The exhaust gas path (8) is shown in the lower section.

During validation, all geometric variables are initially adapted and validated using unfired operating points. After matching the pressure curves at different engine speeds, injection and ignition are added. Due to the intake manifold injection in combination with a gaseous fuel, the mixture formation is very homogeneous, so that the mixture distribution can be well represented by the 1D simulation. The ignition timing as well as the combustion process can be adjusted on the basis of the measured pressure curves in the cylinder. Figure 4 shows a comparison of the measured indexed cylinder pressure with the simulation.

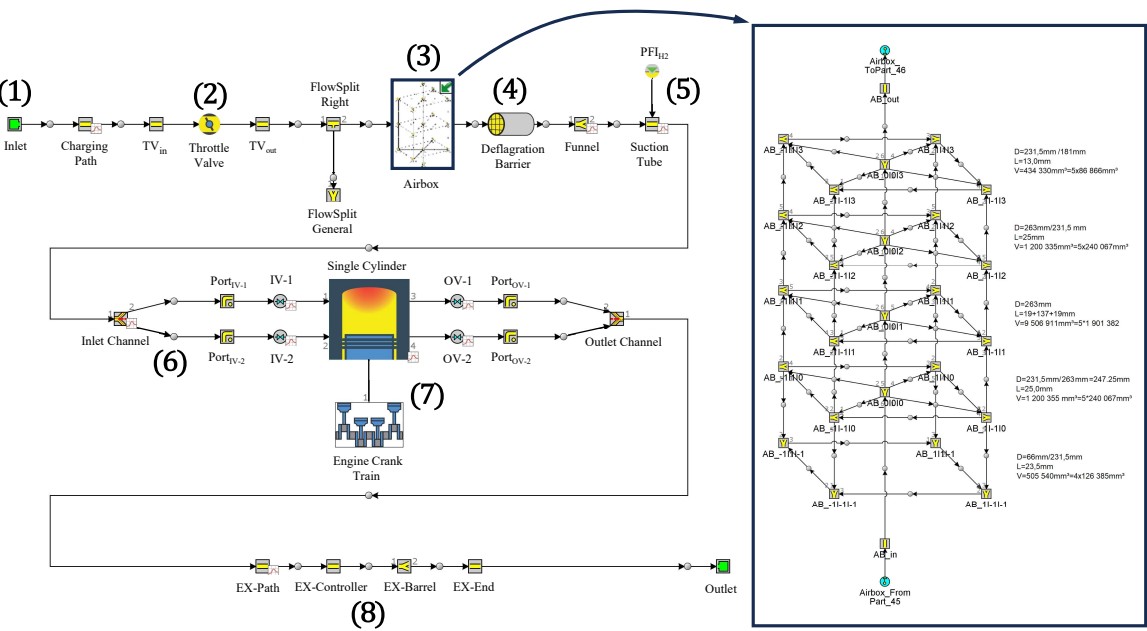

**Figure 3.** 1D-CFD model of the single-cylinder test engine

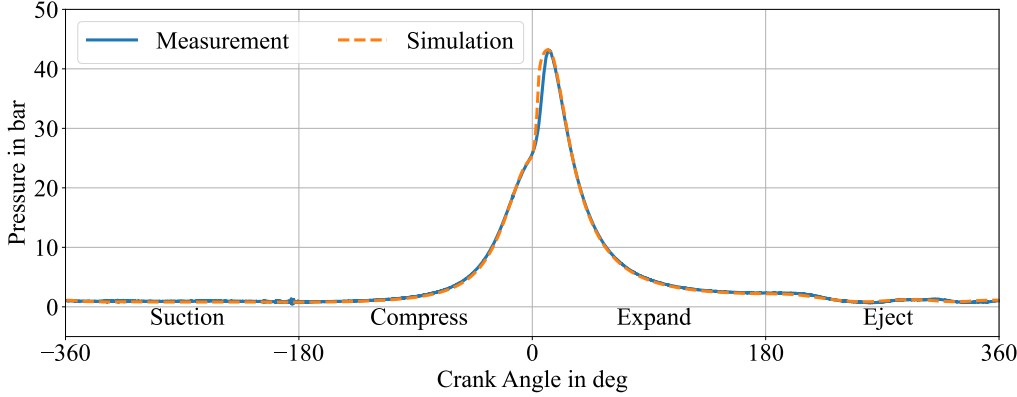

**Figure 4.** Comparison of measured and simulated indicated cylinder pressure

It can be shown that the detailed modeling of the air path provides an excellent simulation of the pressure characteristics during the gas exchange. The pressure curve as well as the maximum pressure are also reproduced well. Only the pressure gradient after ignition shows a slight deviation, but this only has a minor influence on engine performance. The difference is

presumably due to the specific propagation of the flame front, which is essentially dependent on the local mixture composition and geometric boundary conditions. Neither of these is reproduced by the 1D simulation.

The engine model is then extended to a 4-cylinder engine. The geometric parameters of the cylinder are retained, but the valve timing is adapted to the specific requirements. The peripherals are also adapted to the full engine and extended to include a turbocharger and an intercooler. The pipe cross-sections are adjusted to suit the new volume flows.

The engine is designed for an output of $50\,\mathrm{kW}$, whereby this operating point must be capable of continuous operation. The configuration provides for operation with significant lean conditions (lambda > 2) in order to meet the nitrogen oxide emission

regulations even without exhaust gas aftertreatment. The simulation model of the full engine is shown in Figure 5.

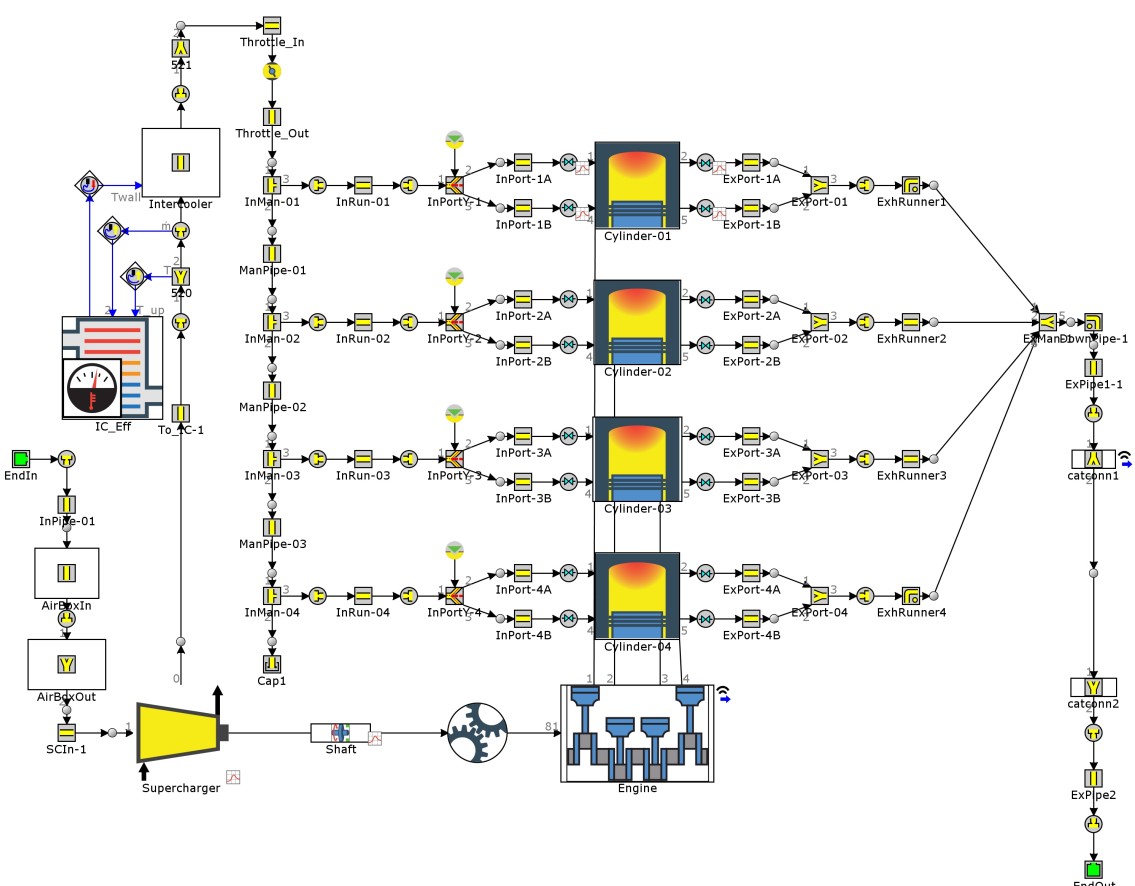

**Figure 5.** 1D-CFD model of the full-engine including compressor and intercooler

A compressor is used in the simulation model to provide the boost pressure. Alternatives such as an exhaust gas turbocharger, which is generally more efficient than the compressor, were also considered. However, due to the low exhaust gas enthalpy during ultra-lean operation, this cannot provide the necessary boost.

As the engine is connected directly to the generator, the speed is constant at 2000 rpm. The power output is regulated by

the air ratio or the degree of turbocharging. A throttle valve is also implemented for the low partial load. In addition, cylinder deactivation in combination with a variable valve train can be used to increase efficiency at low power requirements. The engine application described above results in an indicated efficiency of more than 40% in the operating range above $10\,\mathrm{kW}$ output power. The indicated peak efficiency is 48%.

## 4    Possible system designs for a backup power supply

The first technology that is being considered to supply the platform with power in the event of wind outages is a proton exchange membrane fuel cell (PEM-FC). In addition to the actual fuel cell, other components are required to create a functioning overall system. Such a system is shown with all its essential components in Figure 6.

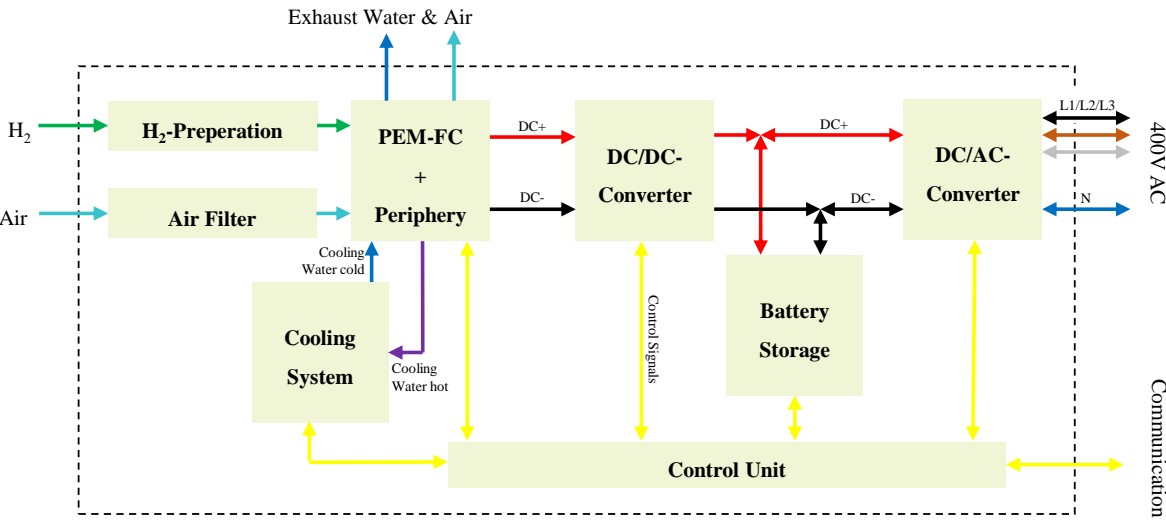

**Figure 6.** Structure of the fuel cell system

At the center of the system is the fuel cell stack with the peripheral components required for operation, such as the air compressor, pumps and valves. In this stack, electrical energy is generated by oxidizing the supplied hydrogen with the help

of atmospheric oxygen. The hydrogen is produced by the electrolyzers on the platform and transported onshore via a pipeline system. This pipeline system is large enough to serve as hydrogen storage for the backup power supply, and no additional tanks are required. However, PEM fuel cells are very sensitive to the purity of the reactants supplied. ISO 14687 allows a maximum contamination of hydrogen with oxygen of 5 ppm for road vehicle applications and 50 ppm for stationary systems (ISO 14687:2019, E). As non of these purity can be guaranteed in the pipeline system, additional treatment of the hydrogen is

necessary. For example, special gas purifiers based on grid-stabilized zeolites can be used for this purpose (Lammertz, 2022). As PEM fuel cells also react very sensitively to salty air, which would result in a sharp drop in performance (Sasank et al.,

2016), the maritime supply air must also be specially filtered. Air filters with an electrostatic pre-filter layer can be used here, which effectively filter salt out of the air. However, these filters only have a maximum service life of one year (Freudenberg Filtration Technologies GmbH & Co. KG, 2024), which corresponds exactly to annual access to the platform for general
maintenance work. With the help of these two measures, the operation of a PEM fuel cell on the offshore system can be ensured, but is associated with increased effort.

An appropriate cooling system is required to dissipate the heat loss generated in the fuel cell to the environment and thus keep it at operating temperature. This system must be relatively large, as stationary low-temperature PEM fuel cells only work with coolant temperatures of around $45\,^{\circ}\text{C}$, which means that there is only a small temperature difference to the environment
on a warm day. On cold days, however, the fuel cell must first be brought up to operating temperature by operating at reduced power before it can be loaded to its maximum power. To close this gap, a battery storage is essential. With an appropriate design, it is also capable of bridging short wind outages without the use of the fuel cell and softening dynamic load changes, which has a positive effect on the service life of the fuel cell. If the turbine stops, the battery storage first takes over the supply of the platform until the State of Charge (SoC) falls below 50%. At this point, the fuel cell is started and brought up to operating
temperature. The fuel cell then takes over the supply to the platform. This ensures that there is always enough energy in the battery storage system for the start-up phase of the fuel cell and any emergency reserves (20% SoC).

In order to adapt the comparatively low and load-dependent output voltage of the fuel cell stack to the higher voltage level of the battery storage system, a DC/DC-converter is installed between the two systems. As the wind turbine and hydrogen production are supplied via a $400\,\text{V}$ three-phase grid, a corresponding inverter is also required to connect the entire system to
the grid. All components are coordinated and monitored by a control unit, which can also be used to communicate with the environment and manage the entire unit. Table 3 shows the most important specifications of the individual components, which are based on products available on the market and have been selected so that the requirements of the system can be met.

**Table 3.** Specifications of the fuel cell system

| Component | Specification | Value |
|---|---|---|
| Fuel Cell | Nominal Continuous Power (BoL; 20% reduction over lifetime) | $68.4\,\text{kW}$ |
| | Minimal Continuous Power | $12.6\,\text{kW}$ |
| | Number of Cells in row | 144 |
| Battery Storage | Capacity (BoL; 30% Reduction over Lifetime) | $56\,\text{kW h}$ |
| | Maximum Charge/Discharge Rate | 1C |
| Cooling System | Maximum Cooling Capacity | $108\,\text{kW}$ |

The second technology that was examined in more detail to supply the wind turbine and the platform with electricity is a hydrogen-powered combustion engine in combination with a synchronous generator. Since hydrogen combustion engines are
a very new technology and accordingly very few systems are available on the market, the unit considered here was designed in detail in section 3. A synchronous generator is used to convert the mechanical power generated by the combustion engine into electrical power. The combustion engine is started via a starter motor with the appropriate battery. The generator can then

be synchronized with the platform's board net. As soon as this is done, the generator can be connected to the electrical system of the platform. In this way, there is no need for additional power electronics between the generator and the platform. The cooling system and the control unit are required for this system in the same way as for the fuel cell system. However, the cooling system can be smaller than in the fuel cell system, as combustion engines operate at a higher temperature level and a large proportion of the heat loss is dissipated via the exhaust gases, so that around half of the cooling surface required for the H2-FC+BS system is sufficient here. (Mayr et al., 2021) H2-ICE is also powered by hydrogen in the pipeline network. As combustion engines are much more resistant to impurities in the hydrogen, especially oxygen, there is no need for additional treatment. The combustion engine also reacts less sensitively to maritime conditions with the air supplied, so that simple air filters, as are common in such applications, are sufficient here.

As a third variant considered here, the H2-ICE system can optionally be extended with a battery storage unit, which can then be connected directly to the platform's grid via an inverter. Similar to the fuel cell system, this can bridge short wind outages completely without the combustion engine having to work. This reduces the operating time of the combustion engine and the hydrogen consumption of the unit. As no ICE heating is required, battery storage can be discharged to 20% SoC (emergency reserve) before ICE has to start. The other functions of the H2-ICE+BS system remain unchanged compared to the H2-ICE system. For the integration of a battery storage in the H2-ICE system, DC-coupled variants such as the fuel cell system are also possible, but these are then associated with increased power electronics complexity.

Figure 7 shows the schematic structure of such a system. The optional battery storage is shown semi-transparent. Table 4 shows the main specifications of the systems.

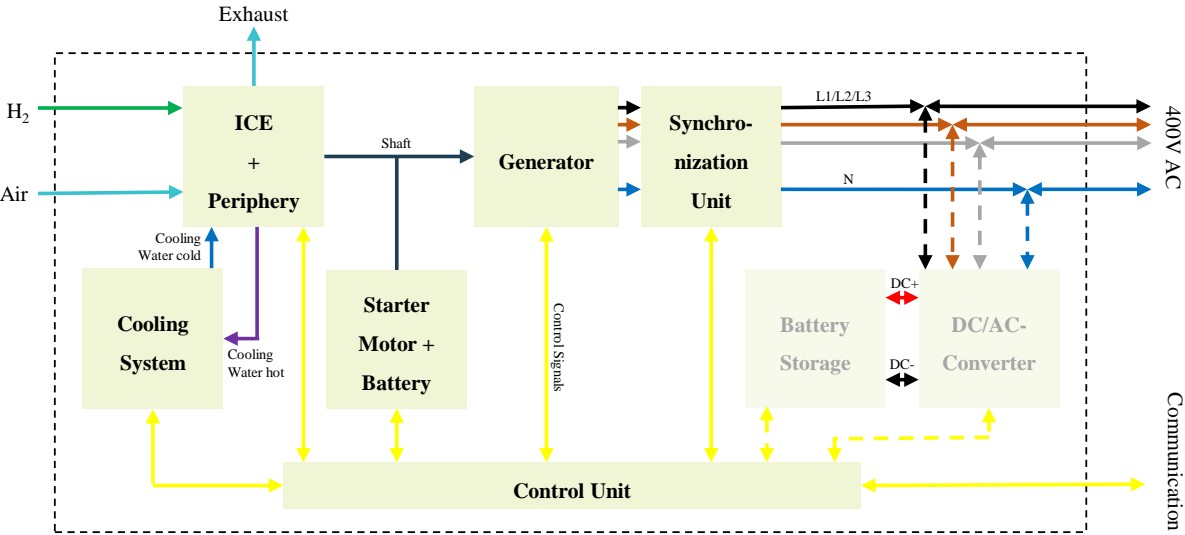

**Figure 7.** Structure of the hydrogen internal combustion engine system with optional battery storage

**Table 4.** Specifications of the internal combustion engine systems

| Component | Specification | Value |
|---|---|---|
| Internal Combustion Engine | Nominal Power | 50 kW |
| Generator | Nominal Power | 67.5 kV A |
| Battery Storage | Capacity (BoL; 30% Reduction over Lifetime) | 56 kW h |
| | Maximum Charge/Discharge Rate | 1C |
| Cooling System | Maximum Cooling Capacity | 54 kW |

## 5 Results

For each of the design options mentioned in section 4, a corresponding dynamic one-dimensional simulation model based on Modelica was created that reflects the behavior of the individual components and the overall system. These allow both the simulation of stationary states and complete cycles from turbine standstill to their restart under various conditions. The Dymola simulation environment and various libraries were used. The models of the individual components are based on a self-developed vehicle simulation which is presented in Burgert et al. (2019) . This was further developed for the new purpose and the components adapted accordingly.

**Efficiency**

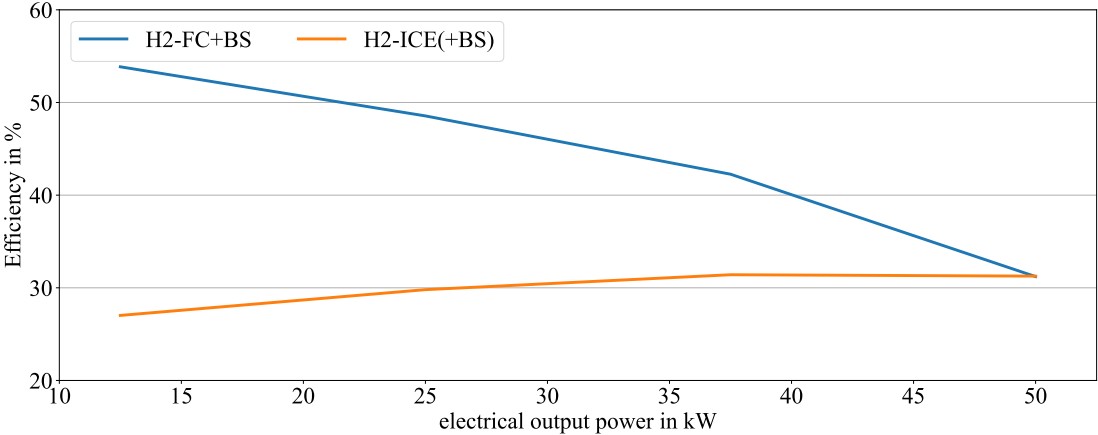

**Figure 8.** Efficiency of the overall systems in steady state

Figure 8 shows the efficiency of the H2-FC+BS system and the H2-ICE(+BS) system. For this purpose, the systems were simulated in a very long standstill scenario and only stationary end state is considered. Four working points with the corresponding power output (12.5 kW; 25 kW; 37.5 kW; 50 kW) and the outside temperature (2 °C; −5 °C; −13 °C; −20 °C) were

simulated for each system. The energy converters are in thermal equilibrium accordingly and the battery storage, if present, is neither charged nor discharged, so the results for H2-ICE and H2-ICE+BS are identical. All internal consumers, especially the cooling system, are taken into account. The results therefore correspond to the efficiency of the energy conversion chain from hydrogen to electrical energy for the platform. The coolers are twice as large in H2-FC+BS system because the operating temperature is lower and significantly less heat is released via the exhaust gases. As a result, the pumps and fans in the H2-FC+BS system require approximately twice as much energy. It can be seen that the efficiency of the H2-FC+BS system decreases with increasing output power. At the lowest output power of $12.5\,\text{kW}$, the H2-FC+BS system reaches its maximum efficiency of 54%. This drops to 31% at the maximum output power of the entire system of $50\,\text{kW}$. The H2-ICE(+BS) system shows an opposite trend. Here, the efficiency increases from 27% at 12.5 kW to 31% at $50\,\text{kW}$ output power, meaning that the two systems are equally efficient at maximum output power. However, the H2-FC+BS system is significantly more efficient at lower power outputs. For example, the fuel cell consumes only $0.696\,\text{kg}\,\text{h}^{-1}$ of hydrogen to provide the electrical output power of $12.5\,\text{kW}$. The combustion engine, on the other hand, requires $1.389\,\text{kg}\,\text{h}^{-1}$. With a power requirement of $50\,\text{kW}$, however, both systems require around $4.8\,\text{kg}\,\text{h}^{-1}$ of hydrogen.

**Hydrogen Consumption**

Based on the environmental conditions analyzed in section 2 and the derived comparative scenarios, the annual hydrogen consumption of the units was quantified. For this purpose, each system was simulated in the four scenarios from the standstill of the turbine and the discharge of the battery storage (if available), through the start and warm-up-process of the energy converters, to the restart of the turbine and the recharging of the battery storage. The resulting hydrogen and electricity consumption was weighted according to the weightings in Table 2. This allows an average consumption per standstill to be quantified and, in combination with the total number of standstills, the annual consumption to be approximated. For the systems with battery storage, not only the directly consumed hydrogen was taken into account, but also the electricity required to recharge the battery storage. For this purpose, the amount of hydrogen that the electrolyzer could theoretically have produced with this amount of electricity was calculated. An electrolyzer efficiency of 70% and a calorific value of the hydrogen of $33.33\,\text{kW}\,\text{h}\,\text{kg}^{-1}$ were assumed. The total hydrogen consumption equivalents of the individual systems are shown in Figure 9 for the analyzed years 2021 and 2022.

As can be seen, the H2-FC+BS system consumes the least hydrogen in both years. In 2021, it would have consumed around 1074 kg of hydrogen. With a hydrogen quantity of $2773\,\text{kg}$ in the same year, the H2-ICE system consumes around 2.6 times as much hydrogen. On the one hand, this is due to the lower efficiency of the system, as can be seen in Figure 8, and on the other hand due to the longer running time of the combustion engine compared to the fuel cell as there is no battery storage. By combining the combustion engine with a comparable battery storage system (H2-ICE+BS), the hydrogen consumption can be significantly reduced. At $1331\,\text{kg}$, this system will only consume around 1.24 times as much hydrogen as the H2-FC+BS system in 2021. As the combustion engine does not need to be warmed up before operation, the battery storage system can be better utilized, which can partially compensate for the significantly lower efficiency of the combustion engine. The consumption

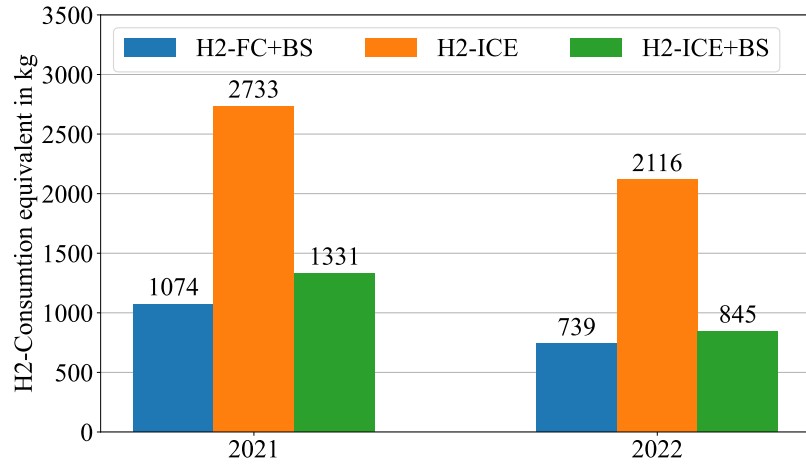

**Figure 9.** Hydrogen consumption equivalent of the different system topologies in 2021 and 2022

ratios for 2022 are very similar, although less hydrogen would generally have been consumed in 2022, which can be attributed to the warmer and windier weather.

**Lifetime**

In addition to the efficiency of the systems and the annual hydrogen consumption, the annual operating times of the critical components can also be estimated on the basis of the analyzed environmental conditions and the simulations carried out. In combination with information on the durability of the systems, the expected service life can be calculated. These are shown in Table 5.

**Table 5.** Estimating the service life of critical components

|  | Operating Time per Year | Specification | Expected Service Life |
|---|---|---|---|
| **Fuel Cell** | 952 h | 20 000 h | 21 years |
| (with Battery Storage) | | (20% Performance Degradation) | |
| **Combustion Engine** | 1950 h | 20 000 h | 10 years |
| (without Battery Storage) | | | |
| **Combustion Engine** | 746 h | 20 000 h | 26 years |
| (with Battery Storage) | | | |
| **Battery Storage** | 167 full cycles | 8000 full cycles | 47 years |
| (with Fuel Cell) | | (100% DoD, 0,5C, 30% Capacity Degradation) | |
| **Battery Storage** | 234 full cycles | 8000 full cycles | 34 years |
| (with Combustion Engine) | | (100% DoD, 0,5C, 30% Capacity Degradation) | |

A service life of around 35 years is assumed for the entire wind turbine. However, the electrolyzer stack has to be replaced after approx. 10 years anyway, which is associated with major maintenance work, meaning that components in the emergency power system could also be replaced if necessary.

Since the battery storage system, both in combination with a fuel cell and an internal combustion engine, has a lifespan roughly equal to that of the entire system, the lifespan of the battery storage system is not critical for the system. In combination with ICE, the battery storage system can be utilized to a greater extent and thus discharged more deeply, since there is no need to retain capacity for the heating phase. This reduces the system's hydrogen consumption, but also reduces the service life of the battery storage system. Therefore, a service life of only 34 years is expected for the battery storage in combination with the ICE. In combination with the fuel cell, on the other hand, a service life of 47 years is expected. Age-related degradation was not taken into account.

A service life of 20,000 operating hours is assumed for both the fuel cell and the hydrogen combustion engine. In combination with the battery storage, these components thus achieve a service life of 21 and 26 years respectively, meaning that the systems would probably have to be replaced once during the life of the wind turbine. If the combustion engine is operated without battery storage, it only achieves a service life of approx. 10 years, meaning that it would always have to be replaced together with the electrolyzer stack.

**Robustness and Maintenance**

In addition to efficiency and the resulting hydrogen consumption as well as the expected service life of the systems, other criteria that are difficult to quantify must also be taken into account for a comprehensive analysis. For example, familiarity with the technologies used has an influence on the regular maintenance of the systems. Although fuel cells have now been tested more intensively for around 20 years, they are still not very widespread and therefore still a very new technology for many. The hydrogen combustion engine, on the other hand, largely corresponds to a conventional combustion engine, which is familiar primarily from the automotive sector and for which there is correspondingly high maintenance capacity. Changes are only made to the ignition and injection system. The reliability of the systems plays a decisive role in an offshore-operated system, as maintenance work in the event of a failure is associated with a great deal of effort and expense. The H2-FC+BS system has significantly fewer mechanical components that are susceptible to wear. Due to the high proportion of electronic components, the system is very easy to monitor remotely so that any damage that occurs can be detected at an early stage. However, the reliability and durability of the fuel cell itself is heavily dependent on the ambient conditions. The conditions on the platform are very unfavorable for a fuel cell, meaning that operation is only possible with complex, expensive and high-maintenance additional systems. An internal combustion engine is much more robust and therefore easier to handle. Research is currently focusing on the influence of hydrogen on lubricants. Furthermore, durability problems are to be expected in the area of the valves, the spark plugs and the injection system. This must be compensated for by regular maintenance and corresponding renewals. Due to the shorter running time of the combustion engine in the H2-ICE+BS system, it can be assumed that the system will require less maintenance and wear parts will need to be replaced less frequently. It is assumed

that maintenance every two years would be sufficient. However, as the system is subject to annual maintenance anyway, this can take place at this time.

**Emissions**

All systems are not critical in terms of local emissions. The fuel cell only emits pure water and the pollutant emissions of the combustion engine can be kept at such a low level through ultra-lean operation that the applicable regulations are complied
even without exhaust gas aftertreatment. Due to the shorter running time of the combustion engine in the H2-ICE+BS system, the total emissions are lower compared to the H2-ICE system.

**Space Consumption**

Due to the limited space on the platform of the wind turbine, the components of the individual systems should be accommodated in a 10ft sea container if possible. As can be seen in Figure 10, there is sufficient space for all three of the considered systems.
At the same time, the components can be arranged in such a way that there is still sufficient space for maintenance work. In addition to the components described in section 4, a switching cabinet is provided in each system in order to connect and wire the systems. This also provides space for the components of the control unit and, in the case of the H2-ICE(+BS), also for the synchronization unit. As the H2-ICE system has the fewest components, it takes up the least space within the container. The H2-FC+BS system and the H2-ICE+BS require approximately the same space within the container. The coolers, which are
about twice as large in the H2-FC+BS as in the H2-ICE(+BS), can be accommodated on the roof of the container in each case. As hydrogen is taken from the pipeline network, no additional hydrogen storage is provided in the containers. If hydrogen in the pipelines runs out because of a very long period of calm, hydrogen would have to be made available from land via the pipelines.

**Costs**

Finally, the investment costs for the unit must also be taken into account. Fuel cells are currently still very expensive. In 2022 the cost of a fuel cell stack was around €800 $\text{kW}^{-1}$ due to the very expensive components and low production capacities (Beyer et al., 2022). In the medium term, a significant price reduction down to €30$\text{kW}^{-1}$ is expected due to increasing demand and the associated higher production capacity. Nevertheless, the fuel cell system still requires expensive additional components for gas treatment and a larger cooling system. The combustion engine, on the other hand, is significantly cheaper to purchase
at around €125$\text{kW}^{-1}$ and requires fewer additional components (Schrank et al., 2021). Only a slight reduction in costs is to be expected here, as hydrogen engines already cost only around 10% - 20% more than a comparable diesel engine. The cost of a stationary battery storage system is around €800$\text{kW}^{-1}$ h (Lichner, 2021). A reduction in costs can also be expected here in the future.

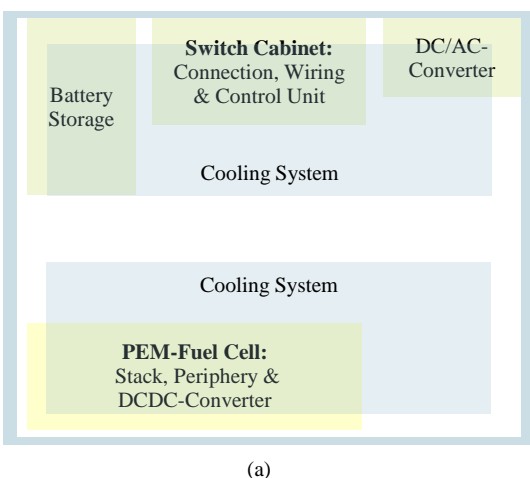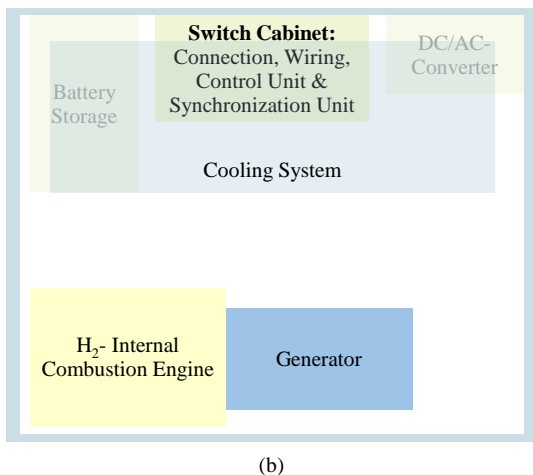

**Figure 10.** Space estimate for the fuel cell system (a) and the H$_2$-ICE-system (b) with optional battery storage

## 6 Discussion

The simulative results are largely based on parameterization using manufacturer specifications. As the components were not available, neither the parameterization nor the individual submodels could be validated. As no corresponding systems were available on the market for hydrogen ICEs, this was redesigned as part of this work. This is based on measurement data recorded on a single-cylinder test bench. Nevertheless, it is not a fully developed and optimized component. Therefore, all quantitative results are subject to a high degree of uncertainty. By setting up and testing such a system, this uncertainty could

be eliminated and the results of the work validated. However, this is not currently planned. The results also only reflect the current state of the art. Development is progressing rapidly both in the field of fuel cells and combustion engines with alternative fuels, so it makes sense to review the results regularly.

The hydrogen consumption is only estimated for the two years under consideration, 2021 and 2022. Although the very different weather conditions in these years lead to very different quantitative results, the qualitative assessment of the systems

is the same despite this. It can therefore be assumed that the assessment will be the same when considering further years.

The simplified consideration of hydrogen consumption with the help of four finite-length scenarios makes it possible to design a sufficiently large battery storage system so that all four scenarios could be completely covered by it and the fuel cell or combustion engine would be superfluous. (Here, an approximate $150\,\mathrm{kWh}$ - $200\,\mathrm{kWh}$ battery would be necessary for Scenario 4). However, since longer standstill times can also occur (up to 3 days in 2021 and 2022), a pure battery storage

system would have to be much larger, which would be very expensive. However, the maximum run-time of such a system would still be limited. The use of hydrogen from the pipeline system for energy supply enables a theoretically infinite supply of energy to the platform, since hydrogen could be supplied from the land. It is also a much more economical alternative.

A large part of the output power of the systems is intended for heating the platform and both the fuel cell and the hydrogen combustion engine generate large amounts of waste heat, which currently has to be dissipated via large radiators. It therefore makes sense to investigate how this waste heat could be used for heating, which would probably be accompanied by a significant increase in efficiency. However, this was not taken into account in the present study, as the integration of the backup power supply on the platform should be as simple as possible. Additional lines and components would be required on the platform and in the other systems in order to use the waste heat.

The backup power supplies (but also the entire platform) were designed in such a way that they can be connected to a standard offshore wind turbine, as the focus of the work was on developing the hydrogen-producing platform. By adapting the electrical connection between the wind turbine and the platform (especially the electrolyzer), the complexity of the power conversion can be reduced, thus saving costs for the overall system. However, the effects of such an adaptation on the backup power supply are comparatively small.

Reliable, qualitative statements about the costs of such systems are very difficult to make. On the one hand, the price of such complex components is constantly changing as development progresses. Secondly, the number of units has a considerable influence on the price. The figures quoted merely reflect literature values.

## 7    Conclusions

The fuel cell is the most efficient system for a permanent backup power supply for the hydrogen-producing wind turbine. However, the fuel cell reacts very sensitively to the environmental conditions of the offshore application, so that its use would only be feasible with significantly increased effort - both technically and financially. The hydrogen combustion engine is much easier to handle for this application due to its robust operation. However, it does entail a significant increase in hydrogen consumption. By using an additional battery storage unit, this additional consumption can be reduced almost to the level of the fuel cell. The hybrid solution consisting of an internal combustion engine used as a range extender for a battery storage system is therefore an optimal solution for the application at hand. This system has the necessary efficiency to avoid wasteful use of the valuable resource hydrogen, but at the same time offers the robustness required for reliable operation in the maritime environment. Table 6 shows the summarized evaluation of the 3 systems under consideration.

**Table 6.** Summarizing evaluation of the individual systems

|  | H2-FC+BS | H2-ICE | H2-ICE+BS |
|---|---|---|---|
| Efficiency in steady state | + | - | - |
| Hydrogen Consumption | + + | - - | + |
| Lifetime | + | - - | + |
| Robustness and Maintenance | - - | + | + + |
| Emissions | + + | + | + |
| Space Consumption | 0 | + + | + |
| Costs | - | + + | 0 |
| **Total** | **+3** | **+1** | **+5** |

*Author contributions.* **Linus Niklaus**: Investigation, Modelling, Parametrization, Writing - original draft, Visualization, Writing - review and editing. **Paul Rothe**: Conceptualization, Investigation, Modelling, Parametrization, Writing - original draft, Visualization, Writing - review and editing. **Johannes Liebertseder**: Conceptualization, Project administration, Writing - original draft. **Martin Doppelbauer**: Conceptualization, Supervisor, Writing - review and editing.

*Competing interests.* The authors declare that they have no known competing financial interests or personal relationships that could have appeared to influence the work reported in this paper

*Acknowledgements.* We acknowledge the funding support of this study by the German Federal Ministry of Education and Research (BMBF) within the project "H2Mare – OffgridWind" under grant number 03HY300D.
These data were collected and made freely available by the BSH marine environmental monitoring network (MARNET), the RAVE project (www.rave-offshore.de), the FINO project (www.fino-offshore.de) and cooperation partners of the BSH. The sea state portal was realized by the RAVE project (Research at alpha ventus), which was funded by the Federal Ministry for Economic Affairs and Climate Action on the basis of a resolution of the German Bundestag.

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
