# Peer review of "Backup power supply for a hydrogen-producing offshore wind turbine - a technology comparison"

_Wind Energy Science, 2024_

## Author Response (AR1)

**Author's response**

Referee Comment

Author Comment (Answer)

Changes in Manuscript (The lines refer to the LaTeX-diff file.)

Referee Comment 1 (10[th] Dec 2024) and Author Comment 1 (18[th] Dec 2024):

A nicely presented and well written paper. A few different options are compared, but the authors are reluctant to make a clear final decision. Hydrogen processing seems to be implemented at the individual turbine level, so it would be interesting to compare against a wind farm approach (offshore energy hub).

Thank you very much for your comment.

I can understand your comments very well, as similar questions arose during the project work. I will try to answer your questions as best I can.

Why implement energy conversion and storage equipment at turbine level and not at farm level? The paper should acknowledge and evaluate the pros and cons of such an approach.

The basic idea of producing the hydrogen at each individual wind turbine was dictated by the project framework. The underlying idea is to create a flexible and modular system based on container solutions that can be assembled on land and then only need to be connected at sea. At the same time, this results in a system that is independent of the overall size of the wind farm. Mass production of the turbines is expected to result in cost advantages. In some places, however, it would certainly be easier to realize one central system instead of many small ones.

**ll. 22-25** -> An explanation of the expected benefits of a platform solution has been added in the introduction.

It seems to be assumed that a standard wind turbine design is being employed, but since the turbine is not grid connected, does this offer additional degrees of freedom, and cost savings?

A "standard wind turbine" from our project partner is assumed. The reason for this decision is to concentrate on the development of the hydrogen production plant within the scope of the project and not to increase the complexity of the new development by redesigning the wind turbine. However, the ulterior motive is certainly to be able to use the same turbine for different applications (with and without hydrogen production) in order to reduce the costs for a single turbine by producing large quantities. By adapting the turbine, however, there is certainly great potential for optimization, especially with regard to the electrical connection of the electrolyser. This would have comparatively little impact on the backup power supply of the plant.

**ll. 307-311** -> The point has been added to the discussion.

Table 1 – include a bigger space between the 2021 and 2022 columns

**Table 1** -> 2mm space between 2021 and 2022 were added.

Line 60 - words such as downtime, outage, interruptions, etc. imply that there is a problem with the wind turbine (which needs to be fixed), while actually it is either too windy or not windy

enough which prevents the wind turbine from operating. Consequently, the terminology used here should be changed.

**ll. 29; 34; 62; 67-78; 89f; 94; 214f; Table 1; Figure 1** -> The wording has been standardized to "standstill time" and "calm wind periods" throughout the document.

Is 2 years of data sufficient to make robust decisions?

The years 2021 and 2022 are two years with very different weather conditions in the German North Sea. 2022 was a very warm and relatively windy year. 2021, on the other hand, was much colder and less windy. These two years therefore represent the extremes for the system. It is to be expected that there will be further years in between. Despite the differences, there is no qualitative difference between the systems in the two years in the efficiency comparison (Figure 9).

**ll. 298-300** -> The point has been added to the discussion.

Fig. 3 – all objects are labelled, but some labels are in German, and some are quite cryptic, so could more user-friendly labels be applied to all objects (in English)?

**Figure 3** -> The graphic has been updated with English labels in a better readable size.

Fig. 4 – change angel to angle

**Figure 4** -> The graphic has been updated with the corrected axis labeling.

Fig. 6 – a legend showing the significance of the different line colours would be helpful

**Figure 6 + 7** -> The graphics has been updated with a labeling of the connections and arrows to indicate the directions of power, signals and mass flow. (See also Referee Comment 2)

Line 135 – maintenance time and access to offshore platform to replace filters?

The maximum maintenance interval for the air filters of the fuel cell is 1 year according to the manufacturer. However, more frequent replacement is recommended to ensure reliability. The aim is to create a system that can operate maintenance-free beyond the annual service interval of the wind turbine in order to avoid additional maintenance time on the wind turbine. However, replacing the filter itself does not require a great deal of time. The problem is the accessibility of the system.

**l. 150f** -> The annual maintenance interval of the platform was added here.

Line 148 – refer to Table 3?

**l. 163** -> The Reference was corrected to Table 3.

Table 3 – how is the specification determined?

Systems available on the market were used to specify the components. These were selected so that they were suitable for this system and could fulfill the required boundary conditions.

**l. 164f** -> The determination of the specification (market available systems to fulfill the system requirements) was added.

Line 168 – some clarification on the nature of the Dymola simulations would be helpful – presumably the simulation was just determining the steady-state efficiency at different loading levels?

Line 183 – as above, the text is rather vague on the nature of the simulations performed

To determine the efficiencies (Figure 8), only the stationary end state of the respective systems was considered. For the consumption calculations (Figure 9), the four scenarios described in Table 2 were simulated dynamically (i.e. from the start of the wind failure and the discharge of the battery storage system, through the start and warm-up process of the energy converters, to the restart of the wind and the recharging of the battery storage system). Both the hydrogen consumed and any electrical energy drawn from the wind turbine were then evaluated.

**ll. 189-191** -> A more detailed description of the simulations has been added in the appropriate places.

Line 245 – cost discussion is very short and very qualitative – can this be made longer and more quantitative in nature?

Reliable, qualitative statements about the costs of such systems are actually very difficult to make. On the one hand, the price of such complex components is constantly changing as development progresses. On the other hand, the number of units has a considerable influence on the price. So it makes a difference whether you build 1 such system or produce it in series. We have therefore deliberately decided against quoting numerical values in order to avoid false statements.

**ll. 280-288** ->The cost analysis was carried out using numerical values and the corresponding sources.

**ll. 312-314** -> The discussion was supplemented to emphasize the difficulty of making quantitative statements about the costs of such systems.

Clarify why waste heat for heating was deliberately left out

The direct use of waste heat was dispensed with as part of the project, as this would have required additional lines and connections on the platform and on the components. The cost of a purely electrical connection, on the other hand, is significantly lower. This should therefore not be implemented, at least in the first generation of the platform.

**ll. 304-306** -> The corresponding declaration was added to the discussion.

What is the final recommendation by the authors? A summary table comparing the different options would be helpful, perhaps with good, average, bad scores for the various criteria outlined in section 5

Our final conclusion is that for the conditions here, a hybrid solution consisting of a hydrogen combustion engine and a battery storage system represents an optimal combination of efficiency and robustness.

**l. 324; Table 6** -> A summarizing table has been added to the conclusion. (See also Referee Comment 2)

I hope I have been able to answer your questions clearly. If you have any further questions, I will be happy to answer them. I will of course take your comments regarding layout, graphics and choice of words into account when revising the document.

Best regards

Linus Niklaus

**General Comments:**

This paper presents a comparative analysis of different backup power supply technologies for hydrogen-producing offshore wind turbines. The study identifies a hybrid system combining a hydrogen combustion engine (H2-ICE) and battery storage as the optimal solution, striking a balance between efficiency, robustness, and reduced maintenance requirements compared to a fuel cell and battery storage system. The work addresses an important challenge in offshore hydrogen production, but some aspects and sections require further elaboration and clarification.

Thank you very much for your comment and the numerous remarks.

I will try to answer your open questions as best I can. I will incorporate your suggestions for improvement into the revision of the paper.

**Specific Comments:**

- The introduction should be expanded to better highlight the motivation and significance of the study. Specifically, it should discuss the consequences of the absence of a backup power supply in offshore hydrogen production and provide an overview of existing approaches to address this issue.

A missing or failed backup power supply in the absence of a grid connection for hydrogen production plant would lead to a total failure of the system, as all processes would come to a standstill and no energy could be provided to restart the system. The backup power supply is therefore of crucial importance for the functioning of the overall system.

**ll. 30-33** ->The introduction has been supplemented by an explanation of the importance of such a system.

- A more detailed discussion of the research gap is needed. Is this study purely a comparative analysis, or does it introduce novel methodologies or findings? Clearly articulating what has not been covered in prior research and how this work contributes to filling that gap would strengthen the paper.

There are already similar comparisons between fuel cell systems and H2 combustion engines in the automotive sector (e.g: Mayr, K., Hofer, F., Ragowsky, G., Gruber, W., Arnberger, A., Kabza, A., Wolf, P., Schmidt, M., and Jörissen, L.: Systemvergleich zwischen Wasserstoffverbrennungsmotor und Brennstoffzelle im schweren Nutzfahrzeug, 2021.). However, under the maritime boundary conditions presented here the comparison is new. And as has been shown, the load profile and the operational conditions on the platform have a decisive influence on the evaluation of such a comparison.

**ll. 39-42** -> The proceeding was supplemented by the intention of the comparison and the focus of previous analog technology comparisons.

- The authors should clearly define the original contribution of this work. Are the system models designed specifically for this study, or have they been adapted from the literature?

The system models are based on vehicle models already available at the Institute (e.g: T. Burgert, A. Dollinger, T. Fischer, "Methodology for Modeling a Plugin Hybrid Electric Vehicle

based on Data Logging," EVS 32, 2019). Their submodels were adapted to the new conditions in order to be able to model the concepts under consideration.

**ll. 192-194** -> A corresponding description of the origin of the simulation models was added at the beginning of the results chapter.

- The purpose of Table 2 and the rationale behind identifying four scenarios should be explicitly explained. How are these scenarios utilized in the subsequent analysis?

The definition of the four scenarios is later used to estimate the annual hydrogen consumption of the systems. The different weightings are used to represent the annual average, which means that it was not necessary to simulate an entire year, as this would have been very computationally intensive.

**ll. 86-89** -> A corresponding explanation of the purpose of the 4 scenarios has been added.

- Figures 6 and 7 should include a clear legend and indicate the direction of power, data, or mass flow to enhance comprehension.

We will add this to ensure a better understanding.

**Figure 6 + 7** -> The graphics has been updated with a labeling of the connections and arrows to indicate the directions of power, signals and mass flow. (See also Referee Comment 1)

- The efficiency determination method in Section 4 requires further clarification. Was the efficiency obtained from literature or experimental data? If the latter, appropriate references should be cited. Also, how does the efficiency curve for H2-ICE without a battery will look?

The efficiencies are derived from the evaluation of the simulation models in a stationary state. Both the primary energy converters (fuel cell and H2 combustion engine) and all auxiliary consumers (e.g. the cooling system or the control system) of the unit and losses in further energy conversion are taken into account. As the system is in a stationary state for this analysis - i.e. the battery storage is not charged or discharged - it makes no difference whether it is installed in the H2-ICE system or not.

**ll. 196-199** -> The method for determining the efficiencies was explained further and it was pointed out once again that the battery storage system is irrelevant in this consideration.

- A comparative table summarizing the key findings for the three technological options analyzed in this study should be added to the discussion section for better clarity.

We will add a table summarizing the results to increase clarity.

**l. 324; Table 6** -> A summarizing table has been added to the conclusion. (See also Referee Comment 1)

- The paper states that three technological options are compared: $H_2$-FC+BS, $H_2$-ICE, and $H_2$-ICE+BS. However, the comparison between the last two options is not clearly presented, making it difficult to discern the additional benefits that battery storage brings to the $H_2$-ICE system. The analysis and discussion of the $H_2$-ICE + BS system are insufficient, despite it being concluded as one of the best options. A more detailed and structured comparison is needed to justify this conclusion.

The H2-ICE+BS system is an extension of the H2-ICE system with the battery storage of the H2-FC+BS system. As the functionality of the two individual systems (H2-ICE and battery storage) is

already described in the respective overall systems, a detailed description is not provided again. The additional battery storage system is not relevant for some of the comparative variables considered (e.g. efficiency and maintenance costs). The biggest difference between the H2-ICE and the H2-ICE+BS systems lies in the annual hydrogen consumption, as can be seen in Figure 9. I will try to better emphasize the effect of the additional battery storage at the relevant points.

**ll. 179-187** -> The differences and similarities between the H2-ICE+BS system and the H2-ICE system were explained.

**ll. 260f; 267f; 271-278** ->In the results chapter, the differences due to the optional battery storage were explicitly addressed in the sections Robustness and Maintenance, Emissions and Space Consumption.

I hope I have been able to answer your questions clearly and clarify any ambiguities. If you have any further questions, I will be happy to answer them.

Best regards

Linus Niklaus

---

## Author Response (AR2)

**Author's response 2**

Report Comment

Changes in Manuscript (The lines refer to the LaTeX-diff file.)

Report Comment 1 (22$^{nd}$ April 2025):

-

-

Report Comment 2 (27$^{th}$ May 2025):

The manuscript presents a comparative study on three backup power supply systems (H2-FC+BS, H2-ICE, H2-ICE+BS) for hydrogen-production offshore wind turbine, based on historical wind speed and temperature data in the German Bight. However, several important issues in the methodology and clarity of technical assumptions need to be addressed before the work can be considered for publication.

1. Technical Assumptions and References:

• In lines 50–55, the authors assume a constant base load of 7 kW and a total power requirement of 50 kW. However, the rationale behind choosing these specific values is not provided. Are they based on empirical data, design guidelines, or previous studies? Proper citations are necessary to justify these assumptions.

**ll. 56-58** -> An explanation of the origin of the assumed load profile has been added.

• The treatment of hydrogen storage capacity is unclear. For example, Fig. 10 does not appear to include any estimation of the space required for hydrogen storage. Since the hydrogen storage capacity critically affects the operational duration of both the FC and ICE systems during standstill periods, this omission introduces uncertainty into the subsequent analysis.

**ll. 140-142 + l. 178** -> In the system descriptions, it was explained that the required hydrogen is taken from the pipeline network, so that no additional storage is planned.

**ll. 291-293** -> In the results section "Space consumption", it was pointed out that no hydrogen storage facilities are planned, as the hydrogen is taken from the pipelines.

• The standstill time is simplified into two scenarios (20 min and 6 h). However, this simplification raises a valid alternative: could a 300 kWh battery (i.e., 50 kW × 6 h) serve the same function? The manuscript should discuss this possibility and justify the advantage of hydrogen-based systems over a purely battery-based solution for this use case.

**ll. 316-322** -> The point was included in the discussion and the scenario described was explained there. It is explained why a pure battery storage system is not expedient, although it would theoretically be possible because of the simplified consideration. The advantages of using hydrogen are highlighted.

2. Control Strategy of Hybrid Systems (H2-FC+BS and H2-ICE+BS):

• For hybrid systems involving battery storage (BS), more detailed discussion on control coordination between FC/ICE and BS is essential. The authors correctly mention (lines 150–155) that a battery is required for FCs to reach operating temperature before full load. However, there is no comparable explanation for the ICE+BS system.

**ll. 158-161** -> The control strategy of the fuel cell system was explained in more detail.

**ll. 185f** -> The start point of the combustion engine is described in more detail.

• As a result, it is difficult to interpret the results shown in Table 5, particularly the higher battery cycling count in the ICE+BS configuration compared to FC+BS. Is this due to differences in startup characteristics, control logic, or load-following behavior? These aspects need further elaboration.

**ll. 246-251** -> An explanation of the different life expectancies of the battery storage system in combination with the fuel cell and the combustion engine has been added.

3. Efficiency Data and Inconsistencies:

• The efficiency curve presented in Fig. 8 lacks explanation. How was this curve derived? Additionally, there is an inconsistency regarding the thermal management of FC vs. ICE. The authors note (lines 165–170) that ICEs require smaller cooling systems due to their higher operating temperatures. Yet, in lines 195–200, it is stated that both FC and ICE systems exhibit the same efficiency at 50 kW. This raises questions about whether the different thermal characteristics have been adequately considered in the performance comparison.

**ll. 200 - 202** -> The explanation of the calculation of the efficiency curves has been detailed.

**ll. 177f** -> The effects of the higher operating temperature and the higher exhaust gas heat flow on the cooling system were described in more detail.

**ll. 204-207** -> The different power consumption of the cooling systems has been detailed.